# Selection of Salinity-Adapted Endorhizal Fungal Consortia from Two Inoculum Sources and Six Halophyte Plants

**DOI:** 10.3390/jof9090893

**Published:** 2023-08-31

**Authors:** Jesús Adrián Barajas González, Rogelio Carrillo-González, Ma. del Carmen Angeles González-Chávez, Eduardo Chimal Sánchez, Daniel Tapia Maruri

**Affiliations:** 1Programa en Edafología, Colegio de Postgraduados, Carr. México-Texcoco km 36.5, Montecillo, Texcoco 56264, Mexico; barajas.jesus@colpos.mx (J.A.B.G.); crogelio@colpos.mx (R.C.-G.); 2Unidad de Investigación en Ecología Vegetal, Facultad de Estudios Superiores Zaragoza UNAM Campus 1, Mexico City 09230, Mexico; 3Centro de Productos Bióticos, Instituto Politécnico Nacional, Yautepec 62739, Mexico; dmaruri@ipn.mx

**Keywords:** microbial inoculant, inoculum infectivity, inoculum effectivity, arbuscular mycorrhizal fungi, dark septate fungi, inoculum adapted to salinity

## Abstract

Soil salinity is a limiting factor in crop productivity. Inoculating crops with microorganisms adapted to salt stress is an alternative to increasing plant salinity tolerance. Few studies have simultaneously propagated arbuscular mycorrhizal fungi (AMF) and dark septate fungi (DSF) using different sources of native inoculum from halophyte plants and evaluated their effectiveness. In alfalfa plants as trap culture, this study assessed the infectivity of 38 microbial consortia native from rhizosphere soil (19) or roots (19) from six halophyte plants, as well as their effectiveness in mitigating salinity stress. Inoculation with soil resulted in 26–56% colonization by AMF and 12–32% by DSF. Root inoculation produced 10–56% and 8–24% colonization by AMF and DSF, respectively. There was no difference in the number of spores of AMF produced with both inoculum types. The effective consortia were selected based on low Na but high P and K shoot concentrations that are variable and are relevant for plant nutrition and salt stress mitigation. This microbial consortia selection may be a novel and applicable model, which would allow the production of native microbial inoculants adapted to salinity to diminish the harmful effects of salinity stress in glycophyte plants in the context of sustainable agriculture.

## 1. Introduction

Soil salinization reduces cultivable land areas and adversely affects agricultural productivity, biodiversity, and water quality, particularly in arid and semi-arid regions. Worldwide, 932.2 million hectares have salinization problems, and most crops of economic interest are not tolerant to salinity, which decreases yields by 20% to 50% [1]. Halophyte plants that naturally grow in saline environments contain many microorganisms in their roots that could be useful for inoculating salinity non-tolerant glycophyte plants. These microorganisms can promote plant growth, influence plant physiology and metabolism, and increase plant tolerance to salt stress.

Arbuscular mycorrhizal fungi (AMF) and dark septate fungi (DSF) are essential components of the soil microbial community. AMF establish mutualistic symbiosis with the roots of 80% of terrestrial plants [2]. These fungi, from the Glomeromycota phylum, are considered obligate biotrophs because they cannot complete their reproductive cycle without an alive host plant. AMF trigger several biochemical and molecular mechanisms in the plant to mitigate salt stress [3,4], including nutrient absorption, hormonal signaling, growth, and the production of regulatory osmolytes. It has been suggested that AMF act as the first selective barrier for ions that can be toxic in saline soils at high concentrations (Na and Cl). Ion discrimination could occur during fungal nutrient absorption from the soil or transfer to the host plant [5].

DSF are a diverse anamorphic group from the Ascomycota phylum. DSF are endophytic microorganisms that colonize plant roots but also have a saprotrophic habit having a marked proportion of saprotrophic genes in their genomes [6]. DSF are found from the tropics to the artic and alpine habitats [7]. These fungi are highly abundant in extreme environments such as saline soils [7]. Their ubiquity in colonizing root plants and observations showing inter- and intracellular interphases suggest that they have a biotrophic nutrition [8]. These are characterized by microsclerotia, a brown to black mycelium, with septate and melanized hyphae in the root’s tissues intercellular and intracellular spaces [9,10,11]. Different to AMF, DSF can be reproduced in vitro. DSF help the host plant to tolerate salt stress, promote plant growth, and facilitate nutrient absorption such as N, K, Ca, Fe, Ni, and Mg [10,12].

Gong et al. [13] referred to the simultaneous root colonization by DSF and AMF as a widespread condition in the plant kingdom. These endorhizal fungi establish functional mutualist associations. However, depending on environmental conditions, the association in AMF may vary from strong mutualism to parasitism [14], but DSF–plant associations range from antagonism to by-product mutualism [6]. It refers to “one organism receiving benefits that another organism produces as a by-product of its self-serving traits”. These authors mentioned that DSF may be considered an intermediate group or transitional form between free-living saprotrophic fungi and mycorrhizal fungi. Plants benefit from DSF more than free saprotrophs but less than AMF. Therefore, the relationship between DSF plants is an example of the evolution of cooperation. In this context, AMF–DSF often co-colonizes the same plant, especially in extreme environments [6].

The coexistence of AMF and DSF in terrestrial plants has been observed in several regions and plants [15,16,17,18,19]. Although the co-colonization of AMF and DSF very often occurs, the significance of the interactions between these fungi and the host in different environmental conditions still needs to be fully elucidated [20]. Under normal conditions, AMF and DSF improve plant growth, P nutrition, antioxidant physiology, and soil nutrient cycling [13]. Xie et al. [21] mentioned that combined plant inoculation with AMF and DSF has more beneficial effects on plant growth and soil properties than non-inoculated or single-inoculated plants. Synergic interactions between these endorhizal fungi were also observed when a plant is under stress, for example, heavy metal contamination [22] and drought [13]. Gong et al. [13] observed two different responses to the interaction of these AMF and DSF under drought stress. A synergic effect of *Rhizophagus irregularis* and *Exophiala pisciphila* caused less membrane electrolyte leakage and oxidative damage of maize leaves, increased photosynthesis, and increased expression of aquaporin gene either in maize or AMF. The co-inoculation in corn plants also had competitive effects on mycorrhizal colonization, maize growth, and root hydraulic conductivity. However, there is limited research on the interaction between AMF–DSF under salinity conditions [23].

As AMF and DSF are essential components in sustainable agriculture, particularly for soils under stressed conditions, they can also be helpful under high salinity [3]. Large-scale production of AMF and DSF inocula is necessary for applying these fungi in agriculture. Traditionally, the most commonly used techniques to isolate and produce large-scale AMF-based inoculum are aeroponics, hydroponics, inert substrate, and in vitro culture [24]; these methods focus on the propagation of mainly one or two species. However, inoculating plants with AMF consortia is more effective than inoculating just one or two species. Regarding DSF production, in vitro propagation and subsequent crop inoculation is the only technique that allows using these fungi as crop inoculants.

Selecting native microorganisms highly adapted to the local conditions, such as salinity, is essential for producing of microbial inoculants; this has been shown for AMF inoculants [25,26]. However, including DSF in crop inoculants is still a unexplored alternative. The present research selected native microorganisms for inoculant isolation and propagation, considering both AMF and DSF native consortia from halophyte plants using two types of crude inoculum (rhizospheric soil and roots). This procedure would offer more options for microorganisms adapted to saline conditions to assist their host plants and mitigate the damaging effects of salinity more efficiently. Moreover, during inoculum fungal screening, physiological criteria based on plant effects under soil salinity conditions should be innovative traits to obtain efficient endorhizal inoculum beneficial for sustainable agriculture facing salinity, climate change, and food safety.

In general, the selection of AMF-based inoculant production protocols focuses on infectivity criteria, considering direct indicators such as species composition, spore density, and mycorrhizal colonization. Indirect indicators, such as the most probable number of propagules and soil microbial biomass can also be considered to assess the quality of a mycorrhizal inoculum [26]. To our knowledge, no studies have evaluated the physiological responses to salinity mitigation in host plants that function as fungal trap cultures and posteriorly use the endorhizal fungi in salinity non-tolerant crops. These responses are related to inoculant effectiveness and may be an additional selection criterion for producing native salinity-adapted inoculants that can colonize roots and efficiently mitigate the deleterious effects of salt stress. Therefore, this research aimed to select microbial inoculants on the base of the intrinsic benefits of the fungi in the context of the stress conditions under which they will be used. In addition, there is limited knowledge about the influence of the type of inoculum, i.e., obtained from the soil or the root, on the production of microbial inoculants. Most studies have isolated the inocula from the soil, which can carry small fragments of colonized roots, spores, and hyphae. Using the roots as the sole source of propagules could also provide potential inoculants for saline soils. Furthermore, DSF fungal groups are yet to be included in the microbial inoculants.

The present study had the following goals: (1) to simultaneously isolate AMF and DSF consortia from halophyte plants, (2) to compare the infectivity and effectiveness of these fungal consortia from two inoculum sources (root and soil), and (3) to analyze the mechanisms of salinity tolerance in AMF that mitigate salt toxicity in plants. This study proposes an alternative for selecting native microbial inoculants adapted to high salinity conditions (NaCl), which can be helpful in mitigating the damage caused by salinity stress in sensitive plants. The main results showed that endorhizal fungi, isolated from roots or soil, influenced alfalfa, as a host plant, fresh mass, and the absorption of nutrients such as P and K rather than Na, which is related to better plant nutrition and less salt stress. This research involved two essential fungi, AMF commonly suggested for plant inoculation and the less explored DSF. In this context, a novel microbial consortium screening from halophyte plants should regard these physiological plant responses during trap culture fungal propagation to select effective native endorhizal fungi adapted to salinity and favor plant fitness facing climate change and food safety.

## 2. Materials and Methods

### 2.1. Root and Rhizospheric Soil Sampling of Halophyte Plants

The sampling site is located at the Colegio de Postgraduados in the east of the State of Mexico with geographic coordinates 19.27° N, 98.54° W, and an altitude of approximately 2236 m (Figure 1). The site of study is separated from the active research agricultural environment. It has been kept almost as natural and unused area. It is located in the former Lake of Texcoco and is characterized by natural salinity caused by NaCl deposits that rose to the surface [27]. Systematic sampling was conducted in February 2021, and 38 samples of rhizospheric soil and fine roots of halophyte plants dominant in the area were collected (*Distichlis spicata*, *Cynodon dactylon*, *Eragrostis obtusiflora*, *Suaeda torreyana*, *Kochia scoparia*, and *Baccharis salicifolia*).

### 2.2. Physical and Chemical Analysis of Rhizospheric Soil

The soil samples were dried in the shade and sieved on a stainless steel mesh No 10. In the saturation extract obtained from the rhizospheric soil samples, pH was measured using a pH meter; electrical conductivity (EC) using a conductivity meter; soluble phosphorus (P) using the Olsen P method; soluble Ca and Mg using AAS (Perkin Elmer 31000, Waltham, MA, USA); and K and Na were measured using flame photometry (NOM-021-RECNAT-2000). Exchangeable Na, K, Ca, and Mg were measured. Available phosphorus was extracted with sodium bicarbonate (0.5 M) in a 1:20 soil:sodium bicarbonate solution ratio. P was determined using UV spectrophotometry at 882 nm [28].

### 2.3. Propagation of AMF and DSF Consortia

The propagation of AMF consists of establishing trap cultures to obtain spores, fragments of colonized roots, external mycelium, and sporocarps in a short time, which are sources of inoculum. Propagation of DSFs is generally performed in in vitro cultures to obtain spores. In the present experiment, endorhizal fungi (AMF and DSF) from saline and saline-sodic soils (NaCl) were simultaneously propagated on alfalfa plants. Half of the trap cultures used as inoculum source soil from the rhizosphere of halophyte plants, while the other half used fine roots.

For endorhizal fungi propagation, 19 composite samples were prepared by mixing soils with similar EC from the same halophyte plant species. Table 1 summarizes the composite samples obtained and used to inoculate alfalfa seeds. Similarly, 19 composite samples were prepared from the root of halophyte plants and used to inoculate alfalfa seeds.

Soil samples with different EC were used as soil samples for trap cultures. One hundred grams of the single samples were inoculated on alfalfa seeds. For composite samples, 50 g of each sample (with similar EC and from the same halophyte plant) were mixed manually to obtain homogeneous samples. One hundred grams of each composite sample was used as inoculum to establish the trap culture.

The trap cultures with root inoculum were similarly established, as mentioned before, for the soil. In individual samples, two g of roots from each halophyte plant were weighed. For composite samples, two g of roots of each plant of the same species, obtained from the soil with similar EC, were mixed and homogenized. In these trap cultures, two g of roots in both cases (individual or composite samples) were used to inoculate alfalfa seeds.

### 2.4. Establishment of Fungal Trap Cultures

From the 19 rhizospheric soil samples and 19 root samples, 38 trap cultures were prepared with three replicates (114 experimental units). Each composite soil or root sample was handled as a fungal consortium, and the same sample number was retained. Each experimental unit consisted of a two kg pot. For trap cultures with rhizospheric soil as inoculum, 1.6 kg of sterile silica sand was first placed at the bottom of the pot, followed by a layer of rhizospheric soil (100 g) and sterile sand, which filled the pot. For trap cultures with roots, two kg pots with 1.6 kg of sterile sand were also used, and a two g layer of roots was placed and covered with sterile sand. Control trap cultures were prepared in the same way but without the inoculum. Alfalfa (*Medicago sativa* var. Moapa) was used as a host plant (trap). Seeds were superficially disinfected with 70% (*v*/*v*) alcohol and rinsed thrice with sterile and distilled water. Water excess was removed with sterile filter paper, and one g of seed per pot was sown.

The trap cultures were maintained under greenhouse conditions for 11 months (average temperature 22 °C) at the Colegio de Postgraduados, Campus Montecillo, State of Mexico. The plants were irrigated every third day with a modified Hoagland nutrient solution low in phosphorus (20 µM KH_2_PO_4_), according to Millner and Kitt [33]. The pots were rotated every week in a clockwise direction.

### 2.5. Evaluation and Selection of Endorhizal Fungi Consortia

Plants were harvested 11 months after planting and inoculation. Endorhizal consortia (AMF and DSF) from rhizospheric soil and roots were selected based on their infectivity and effectiveness on alfalfa plants. Infectivity was based on root colonization [34] and the number of spores on the substrate; both variables are normally used for AMF. Root colonization by DSF was considered an additional variable, so dual colonization results are presented. To select effective consortia, it considered the following criteria: high fresh weight production, plant nutrition (considering foliar N and P concentrations), and adequate measures of several physiological variables related to salt stress mitigation, i.e., limited Na absorption (since this cation is toxic) and high absorption of osmoregulatory cations (K, Ca, and Mg).

#### 2.5.1. Fungal Infectivity

Alfalfa roots from the trap cultures were washed and cut into 1 cm segments. Segments were clarified with 3% KOH for 12 days until the removal of root pigmentation. Change of KOH was performed every three days when it was dark. In the last change, KOH was nearly transparent and roots were visually without pigmentation. Root segments were immersed in 3% HCl for 15 min and stained with 0.5% Trypan blue in 50% glycerin for three days, according to Phillips and Hayman [35]. The percentage of total colonization by AMF and DSF was determined by the method described by Koske and Gemma [36]. Permanent preparations of root segments were observed under an optical microscope (40× objective). The frequency of AMF structures was quantified by considering the presence or absence of arbuscules, vesicles, or hyphae. DSF-colonized structures were established as those with microsclerotia and dark septate hyphae in the same root segments.

The frequency of AMF and DSF (%) was calculated with the following formula:$$\%\;Total\;colonization\;=\;\frac{number\;of\;fields\;colonized\;by\;any\;fungal\;structure}{total\;number\;of\;observed\;segments}\;\times 100$$

Spores of AMF from each fungal trap culture were extracted by the wet sieving technique Gerdemann and Nicolson [37]. Spores were observed under a stereoscopic microscope. The number of spores was reported per 20 g of dry substrate.

#### 2.5.2. Taxonomical Morphotypes Identification of Spores of AMF

From the selected infective and efficient fungal consortium, the analysis of morphotypes of AMF was also performed to know their taxonomical name. This was conducted in the original and propagated consortium samples.

#### 2.5.3. Fungal Effectiveness

Alfalfa plants were harvested, and split into the aerial part and the root. Fungal effectiveness was assessed on the aerial part only since roots can be used as a source of inoculum for other plants after evaluating their infectivity (as mentioned previously). The aerial plant material from each pot was weighed and placed separately in paper bags for drying at 60 °C for 72 h. The material was grounded and digested with a mixture of HNO_3_:HClO_4_ (3:1) in an open system. The sample was diluted to 50 mL with water, and the macronutrients analyzed were the following: N, P, K, Ca, Mg, and Na. The concentration of Ca and Mg was determined with an atomic absorption spectrometer [38], while Na and K were quantified using flame photometry [39]. P concentration was obtained using the ammonium molybdate blue method [40]. The foliar percentage of N was analyzed using the Kjeldahl method [41].

#### 2.5.4. Elemental Analysis in AMF Spores and Hyphae

To determine the possible relationship between AMF fungal structures and accumulating Na to decrease its transport to the host plant, samples of spores and hyphae from AMF were analyzed using environmental scanning electron microscopy (ESEM) coupled with EDX. A backscattered electron detector and an X-ray detector (Bruker, Quantax 200, Mannheim, Germany) were used. Elemental analysis using EDX was performed on these fungal structures with an accelerating voltage of 30 kV, a spot size of 700 (arbitrary units), and a counting rate of 1000–9000 counts per second (cps). Spores from Consortia 2, 8, and 11, collected from extremely saline soil, moderately saline soil, and saline soil, respectively, were chosen for this analysis.

### 2.6. Surface Disinfection of Roots, DNA Extraction, and PCR Amplification

A fungal molecular study was conducted on the original consortium, which originated from *D. spicata* and showed the best characteristics of infectivity and effectiveness. The roots of *D. spicata* were separated from the adhered soil, washed with 2% NaCl, and dried with sterile filter paper. Roots were processed with 70% ethanol to disinfect the surface (2 min), washed with sterile 2% NaCl (three times), disinfected with 15% H_2_O_2_ (5 min), and, finally, washed with sterile 2% NaCl (three times). The solutions obtained after the final washing of each sample were evaluated to ensure surface disinfection efficiency using agar plates. Only successfully disinfected root material was used for further analysis. DNA isolation from 20 mg of disinfected root material was performed according to the protocol (plant and fungal DNA purification kit, EURx). Isolated DNA was placed in 1.5 mL Eppendorf tubes and shipped in cold packs (blue ice) to NOVOGEN (Sacramento, CA, USA) for amplicon analysis (Illumina Platform-MiSeq, San Diego, CA, USA). The region ITS1-5F was amplified using primers ITS5-1737_F(5′-GGAAGTAAAAGTGCTAACAAGG-3′) and ITS-2043_R (5′-GCTGCGTTCTTCATCGATGC-3′). The processing of the sequencing data was carried out in the NOVOGEN laboratory with the following methodology: Paired-end reads were assigned to samples based on their unique barcodes and truncated by cutting off the barcode and primer sequences. Paired-end reads were merged using FLASH (V1.2.7), a very fast and accurate analysis tool, which was designed to merge paired-end reads when at least some of the reads overlap the read generated from the opposite end of the same DNA fragment, and the splicing sequences were called raw tags. Quality filtering on the raw tags were performed under specific filtering conditions to obtain high-quality clean tags according to the Qiime (V1.7.0) quality-controlled process. The tags were compared with the reference database (SILVA138 database) using the UCHIME algorithm to detect chimera sequences, which were removed. Then, the effective tags was finally obtained. Sequences analysis were performed with Uparse software (Uparse v7.0.1090) using all the effective tags. Sequences with ≥97% similarity were assigned to the same OTUs. Representative sequence for each OTU was screened for further annotation. For each representative sequence, Qiime (Version 1.7.0) in Mothur method was performed against the SSUrRNA database of SILVA138 Database (for species annotation at each taxonomic rank (threshold: 0.8~1) (kingdom, phylum, class, order, family, genus).

### 2.7. Experimental Design and Statistical Analysis

A completely randomized design with three replicates was used to evaluate the effectiveness and infectivity of fungal consortia on alfalfa plants. The number of composite mixtures represented the established fungal consortia (19), and a control treatment (uninoculated) was included. For all variables, the Shapiro–Wilk normality test was performed, and the homogeneity of variances was corroborated using Bartlett’s test (α = 0.05). All the variables presented normal distribution, so data transformation was not necessary. The data were analyzed with descriptive statistics, ANOVA (*p* ≤ 0.05), and Tukey’s test for comparison of means (*p* ≤ 0.05) using R statistical software version 4.0.5. Multivariate analysis (PCA = principal component analysis) was performed with the R package Factoextra version 4.0.5 [42].

## 3. Results

### 3.1. Physical and Chemical Analysis of the Rhizospheric Soil of Each Composite Sample

The EC, SAR, and soluble cations analyzed in the rhizospheric soil of the study area showed high variability (Table 1). The EC ranged from 0.9 to 43 dS m^−1^. The order of concentration of soluble cations in the soil was Na > K > Ca > Mg. The SAR ranged from 14 to 580 mM. Na concentrations ranged from 12 to 656 mM, and all samples exceeded the normal concentrations for soils. However, the K and Ca concentrations were normal, but the Mg concentrations were low. The available P concentration was between 6.41 and 17.38 mg kg^−1^ (Table 1). The pH ranged from 7.8 to 9.6.

### 3.2. Infectivity of Fungal Consortia

#### 3.2.1. Arbuscular Mycorrhizal Fungi

##### Mycorrhizal Colonization

All roots of alfalfa plants inoculated with soil or root inocula showed characteristic colonization by AMF and DSF structures. AMF formed coiled hyphae, vesicles, and arbuscules on the roots. DSF structures were dark-colored septate hyphae and microsclerotia growing intracellularly in the root cortex (Figure 2).

When the soil was used as the inoculum, the percentage of AMF colonization was between 26% and 56%. Consortia 2, 3, 4, and 6 propagated from the soil, promoting the highest root colonization (Figure 3a). In contrast, the lowest percentage of AMF colonization was detected in plants inoculated with Consortia 13, 16, and 17, which belong to soils with low EC and Na concentrations.

Regarding the plants inoculated with the root-derived consortia as inoculum, the total colonization percentage was between 10% and 56%. Plants inoculated with Consortia 3, 7, and 12 showed the highest colonization percentage, while all other plants had less than 30% (Figure 3a). The average of colonization rate differed between plants inoculated with soil and plants inoculated with roots (43% and 23%, respectively). However, Consortium 7 showed a similar percentage of colonization in alfalfa when using soil or root as inocula. Consortium 12 had a higher percentage of colonization with root inoculum than soil inoculum. All other plant colonization percentages were higher when using soil as inoculum than root (Figure 3a).

Plants inoculated with root consortia 3, 7, and 12, which had the highest percentage of colonization, consisted of *Glomus* sp. and *Diversispora* sp. (Consortium 3), *Rhizophagus* sp. and *R. aggregatus* (Consortium 7), and *Septoglomus* sp. (Consortium 12). These AMF genera native to saline soils presented as infective propagules on alfalfa.

##### Number of Spores of AMF

Spore production depended on the composite sample and inoculum source. The number of spores produced by the consortia using the soil inoculum ranged from 48 to 95 in 20 g of soil (Figure 4). Consortium 1 produced the highest number of spores among all consortia, despite having the highest EC value (42.5 dS m^−1^) and Na concentration in soil (655.9 mM).

In plants inoculated with the root consortia, the range of spore number was between 28 and 174 per 20 g of substrate. The highest number of spores was obtained in plants with Consortia 3 and 7 (Figure 4). To our knowledge, no research had previously used the roots of halophyte plants for inoculum production; this study is the first to show spore production from halophyte plant roots in a trap culture.

When relating inoculum source, spore production in Consortia 3, 7, and 12 was higher when the root was used as inoculum than the soil. In contrast, Consortia 1, 2, 4, 5, 6, 8, and 19 had higher spore numbers when using soil as inoculum compared to the root. However, in Consortia 10, 11, 13, 14, 15, 16, 17, and 18, the number of spores was similar with both inoculum sources.

#### 3.2.2. Dark Septate Fungi

The percentage of DSF colonization from soil consortia was between 12% and 32% (Figure 3b) in the alfalfa roots. Alfalfa roots from Consortia 3, 7, 14, and 16 had the highest percentage of colonization, which differed from the other consortia. In plants inoculated with root consortia, the colonization rate ranged from 8% to 24%. The highest percentage of colonization was observed in Consortia 6, 7, 9, 12, and 13, which was different from the rest of the plants (Figure 3). AMF showed a higher percentage of colonization than DSF and varied according to the fungal source used in the trap culture (Figure 3). The average percentage of AMF colonization in soil consortia was 45% and 22% for DSF; in root consortia, it was 23% for AMF colonization and 13% for DSF colonization. 

### 3.3. Mitigation of Salt Stress in Alfalfa Plants

#### 3.3.1. Effectiveness of Fungal Consortia Propagated on Alfalfa

##### Fresh Weight

Plants inoculated with Consortia 3, 7, 14, 17, and 18, pertaining to the soil, produced the highest fresh weight of the aerial part in alfalfa plants (Figure 4). The response with Consortia 3 and 7 is particularly interesting because the original soils presented high EC. The increase in fresh weight was 20% for Consortium 3 and 41% for Consortium 7, compared to control (non-inoculated) plants. The dry weight of most of the other consortia was similar to that of the control plants despite the high Na concentration and high EC value of the soil in some composite samples containing these consortia.

Regarding using the root as inoculum, plants inoculated with Consortia 15 and 19 had the highest fresh weight in the aerial part of the alfalfa plants (Figure 4). The fresh weight was different in the other inoculated plants, including the control. Alfalfa plants showed higher fresh weight when the soil was used as inoculum compared to the root, except in plants with Consortia 1 and 19 (Figure 5).

##### The Foliar Concentration of N and P

The N content in the foliage of plants inoculated with soil ranged from 1.2% to 4.6%. Plants inoculated with Consortium 11 had the highest percentage of N (Table 1). In contrast, the N content in the control plants was 1.08% (*T. foenum-graecum* a). Consortia 1, 4, and 6 from the sodic soil also influenced typical foliar N percentages. Consortia 1, 4, 6, and 11 mitigated the negative effect on N absorption in alfalfa plants. When root was used as inoculum, the foliar N content had a narrower range (1.17–3.74%) than when the soil was used as inoculum. Again, Consortia 4 and 6, as well as 16, influenced the highest foliar N percentage (3.68%, 3.08%, and 3.74%, respectively). Therefore, these consortia are efficient in N absorption in alfalfa plants. 

Most plants inoculated with consortia from both soil and root increased the foliar concentration of P. The foliar concentration of P was higher in plants inoculated with Consortia 1, 3, and 10 (from the soil). Also, P concentrations differed from the rest of the inoculated plants and the control (Figure 6b). Plants inoculated with Consortia 12 and 14 (soil inoculum) showed foliar P concentration similar to that in control plants. The highest P concentration was observed in Consortia 8, 10, 13, and 14 when roots were used as the inoculum source. Consortium 10 of both inoculum sources promoted the highest foliar P concentration. Consortia 12 and 14 (soil inoculum) showed foliar P concentration similar to control plants.

##### The Foliar Concentration of Na and Protective Osmolytes

The foliar Na concentration of plants inoculated with soil consortia ranged from 168 to 453 mmol kg^−1^ DW (Figure 7a). In control alfalfa plants without soil addition (and therefore, without an external Na source), the foliar concentration was less than 50 mmol kg^−1^. Plants inoculated with Consortia 1, 4, 9, 12, and 15 (Figure 7a) had the lowest leaf Na concentration. Regarding inocula obtained from the roots, inoculated plants had similar leaf Na concentrations to non-inoculated plants, and all plants showed average Na concentrations.

In the present study, leaf K concentration increased significantly in alfalfa plants inoculated with the soil consortia (Figure 7b) except for 15, which had similar concentration as non-inoculated plants (310 mmol kg^−1^ DW). Plants inoculated with Consortia 1, 8, 17, 18, and 19 from soil had the highest leaf K concentration (1016, 1086, 1028, 1028, 888, and 984 mmol kg^−1^ DW, respectively). Regarding the plants inoculated with root as inoculum, all consortia except 1 and 3 increased the foliar concentration of K. Consortia 2, 4, 13, and 14 were the most efficient (Figure 7b). Importantly, all plants with the root consortia as inoculum only received nutrient solution as an external source of K.

In all consortia (from both soil and root), the foliar Ca in alfalfa plants was significantly higher than that of the control plants, which was 270 mmol kg^−1^ DW (Figure 7c). Regarding the plants inoculated with the root consortia, without Na from the soil, Consortium 8 substantially increased the foliar Ca concentration (1102 mmol kg^−1^ DW) compared to control plants (270 mmol kg^−1^ DW).

The native consortia of the two inoculum types increased the leaf Mg concentration in alfalfa (Figure 7d). The range of leaf Mg concentration of alfalfa plants inoculated with the soil-derived fungal consortia was between 113 mmol kg^−1^ DW to 280 mmol kg^−1^ DW. In contrast, control plants had a leaf Mg concentration of 50 mmol kg^−1^ DW (Figure 6d). Consortia 1, 3, 4, 6, 7, 7, 8, 13, 14, 15, 18, and 19 had the highest leaf Mg concentration, although the Mg concentration in soil was lower than the concentration typically found in soil solution (800 mM). The range of leaf Mg concentration in the plants inoculated with the root consortia was between 157 mmol kg^−1^ DW and 368 mmol kg^−1^ DW, with Consortia 8 and 9 promoting the highest leaf Mg concentration compared to the control plants (Figure 7d).

### 3.4. Scanning Electron Microscopy (SEM) and Elemental Analysis in AMF Fungal Structures

Figure 8 shows micrographs obtained by ESEM and the elemental composition of spores and external mycelium of AMF in some trap cultures. The Na concentration in spores was low (0.03–0.29%), but Na was not detected in the external hyphae despite its abundance in the soil of provenance, for example, Consortia 2 and 8.

The EDAX analysis showed that spores had low K concentration (0.13% to 0.15%), while hyphae presented K concentration between 0.12% and 0.28%. Consortia 2 and 8 spores from the soil had 2.57% and 2.59% of Ca, respectively. The external hyphae of Consortia 2 and 8 had 1.69% and 2.15% Ca. In spores from Consortia 2 and 8 (from soil inoculum), 0.96% of Mg was observed, while in Consortium 11 (from root inoculum), it was 0.43%.

Iron, Cu, and Si were also detected in the spores and hyphae of the consortia. Consortia 2 and 8 spores, pertaining to the soil, had 0.88% Si, while Consortium 11, isolated from the root, had 0.17% Si. The external hyphae of Consortia 2 and 8 had 0.09% Si and 0.75% Si, respectively.

### 3.5. Principal Component Analysis (PCA)

The first four components of the PCA explained 65% of the total accumulated variation (Figure 9). PC1 accumulated 25.3% and revealed the most significant variables in the study: the foliar concentration of N, P, and K; the percentage of AMF colonization; the number of spores of AMF; and the soil concentration of P, K, and Na. PC2 explained 15.4% of the variance and was influenced by the foliar concentration of Na, Ca, and Mg and DSF colonization. PC3 accumulated 13.0% of the variation; the related variables were soil Ca concentration, soil Mg concentration, and EC. PC4 accumulated 11.2% of the variance, and alfalfa fresh weight was related. The analysis also showed that the response of the control plants was separated from the effect of the inoculated plants. Consortia 18 and 19 from soil and Consortia 15 and 19 from roots were related to alfalfa fresh weight in this component. Plants inoculated with Consortia 1 and 2 from soil and root were grouped in Component 1.

### 3.6. Selection of Fungal Inoculants

Based on the previous results, Consortium 1 from soil was selected for further use as an inoculant. Soil from this consortium had the highest EC and Na concentration (Table 1). Under these conditions, which might negatively affect fungal infectivity, Consortium 1 had the highest spore production (96 in 20 g of the substrate; Figure 4) and a DSF colonization percentage of 19% (Figure 3b). Soil Consortium 1 was selected for its effectiveness in nutrient absorption under salinity conditions. It had the lowest foliar Na concentration (Figure 6) and increased foliar P and K concentration (Figure 7).

### 3.7. Identification of AMF Spore Morphospecies in the Selected Consortium

Soil Consortium 1 morphospecies before trap culture were *F. mosseae* (67%), *C. claroideum* (13%), *R. irregularis* (12%), and *Diversispora* sp. (8%). Soil Consortium 1 morphospecies after trap culture were *F. mosseae* (47%), *C. claroideum* (21%), *R. irregularis* (21%), and *Diversispora* sp. (11%). Figure 10 shows these fungal morphospecies.

### 3.8. Endorhizal Community of the Selected Consortium

The endophytic fungal community was identified in the roots of *D. spicata*. This plant was one of the dominant halophytes from which Consortium 1 originated. This consortium was selected based on its infectivity and effectiveness.

The analysis of ITS sequences revealed the predominance of two phyla: Ascomycota (59%) and Glomeromycota (8%), as well as 33% of the unidentified group (Figure 11). At the genus level, *Fusarium* (29%) and *Preussia* (5%) were the most abundant in the phyla Ascomycota, while *Rhizophagus* (4%) was in Glomeromycota.

## 4. Discussion of Results

### 4.1. Physical and Chemical Analysis of the Rhizospheric Soil of Each Composite Sample

Several authors have detected the high variability observed in EC, SAR, and soluble cations in the same geographical area. For example, Valenzuela-Encinas et al. [27] reported ranges of CE from 2.3 to 200 dS m^−1^, Castro-Silva et al. [43] from 71 to 159 dS m^−1^, and Beltrán-Hernández et al. [44] from 22 to 150 dS m^−1^. In the present research, the CE (0.9 to 43 dS m^−1^) was not as high as those observed by the mentioned authors. The SAR (12–656 mM) had intermediate values in comparison to these observed by Beltrán Hernández et al. [44] and Santoyo de la Cruz et al. [45] who reported SAR values between 103 and 1718 mM and between 11 and 43 mM, respectively. In the present research, the soil pH was classified as alkaline [46]. Valenzuela-Encinas et al. [27] reported pH ranges between 7.8 and 10.1 in soils from the former lake of Texcoco.

The average normal available P concentration in non-saline soils is 25 mg kg^−1^ soil, so all composite samples from the present research were deficient in P. Similar results were observed by Xie et al. [47], who observed the available P concentration of 12 mg kg^−1^ in saline soils. While Mahmood et al. [48] reported even lower P soil concentration than in the present research of 2.9 mg kg^−1^ and 2.8 mg kg^−1^ of P in two saline soils with EC of 6.59 dS m^−1^ and 4.21 dS m^−1^, respectively.

### 4.2. Infectivity of Fungal Consortia

#### 4.2.1. Arbuscular Mycorrhizal Fungi

##### Mycorrhizal Colonization

The results showed that both inoculum sources, roots and soil, could generate infective consortia of AMF and DSF. The fungal structures observed in alfalfa roots were typical for AMF, and those colonized by DSF were characteristic of the genera *Ammopiptanthus* [49] and *Paraphoma* [50]. The literature shows a limited evaluation of AMF infectivity considering different inocula sources (soil or root) under saline conditions. This is particularly interesting because colonization above 50% is regarded as a desirable infectivity parameter for AMF inocula [34], and these soils had high Na and EC concentrations (Table 1). Therefore, the infectivity of the consortia supports their use as prospects for inocula. INVAM [34] showed that commercial inocula (from non-saline sites) have root colonization between 19% and 54%, comparable to that obtained under salinity conditions in the present investigation.

Particularly, seven consortia promoted colonization greater than 50% in alfalfa roots under high salinity conditions, four consortia from the soil, and three consortia from roots as a source of inoculum. The use of roots as inoculum is not common; however, it represents a viable source of propagules (spores, hyphae, vesicles) that can colonize plant roots, but this depends on the fungal species in the roots [51]. For example, roots colonized with *Glomus* and *Acaulospora* may be infective, but not so when colonized by *Scutellospora* and *Gigaspora* [51,52].

##### Number of Spores

The maximum number of spores propagated in alfalfa with soil as inoculum was 95 in 20 g of soil, while with roots, it was 174. Bencherif et al. [26] observed higher spore numbers using soil from halophyte plants to produce inoculum of AMF. These authors quantified a maximum of 650 spores in 10 g of soil in alfalfa after 24 months of trap culture with aerial part cutting and reseeding every four months. In the present research, the trap culture lasted 11 months, and alfalfa plants were uncut. Using soils with natural salinity (not anthropogenic), Aliasgharzadeh et al. [53] reported 160 spores in 20 g of soil in alfalfa plants with EC of 12.2 dS m^−1^, 288 spores in onion with 7.3 dS m^−1^, 239 spores in wheat with 12.1 dS m^−1^, and 230 spores in barley with 21.1 dS m^−1^. However, in other studies, the number of spores has been lower [54], and no spores have been found in the rhizospheric soil of halophyte plants with EC higher than 45 dS m^−1^ [55,56]. Although the number of propagules in an AMF inoculum is related to its infectivity and then inoculum quality, there is no concrete data to establish the grade of infectivity of inocula based on the number of spores. Under trap culture conditions, some species sporulate readily, while others infrequently or not at all [57,58]. However, this depends on the ability of AMF species to sporulate under the conditions that fungi and plants establish.

#### 4.2.2. Dark Septate Fungi

With both kinds of inoculum sources, AMF and DSF colonized alfalfa roots. These results show, for the first time, that DSF native to salty soils can colonize the roots of alfalfa plants along with AMF when using natural inocula. In the present research, the maximum colonization in alfalfa roots by DSF was up to 32%. Operationally, DSF is in vitro isolated and then inoculated in plants [59]. Under these experimental conditions, González Mateu et al. [60] inoculated a native DSF consortium from saline soil into native and invasive lineages of the plant *Phragmites australis*. They observed 61% colonization in the native and 57% in the invasive lineage. Farias et al. [61] separately inoculated the DSF *Sordariomycetes* sp. and *Melanconiales elegans* in cowpea*s*, both isolated from the halophyte plant *Vochysia divergens*. These authors observed that plants inoculated with *M. elegans* had a maximum colonization rate of 74%, while the colonization of *Sordariomycetes* sp. was 70%. The percentages of colonization by DSF discussed previously are higher than those of the present investigation, both in soil or root inoculum. This difference may be due to the low initial infectivity of the native fungal propagules, the fact that some DSF are host-specific, as mentioned by Hawksworth and Rossman [62], or due to multiple factors, such as climate, soil type, and salinity concentration, that may influence colonization.

AMF had a higher percentage of colonization than DSF. Fuchs and Hasewandter [16] indicated that AMF colonization does not always predominate over DSF colonization and that the colonization pattern is host–plant dependent. Cofré et al. [23] evaluated the percentage of AMF and DSF colonization in roots of *Atriplex cordobensis* in soils with different EC (2.0, 4.4, and 19.7 dS m^−1^). The authors observed similar colonization by AMF and DSF in the soils with the lowest EC values, and colonization by DSF was higher than AMF in the soil with high EC. Sonjak et al. [63] observed higher colonization by AMF than DSF in 12 halophyte plants growing naturally in saline soils. As mentioned before, multiple factors may influence the percentage of colonization of both types of fungi; however, this variable is relevant to infectivity or quality inoculum.

### 4.3. Mitigation of Salt Stress in Alfalfa Plants

The effectiveness of fungal consortia on the response of alfalfa plants includes the participation of both types of fungi (AMF and DSF). Since it is impossible to separate the individual effects, the discussion will be inferred from the current knowledge of each type. In general, the results show that the response to inoculation depends on the type of inoculum (root or soil), consortium, and physical or chemical characteristics of the original soil.

Unlike AMF, DSF can be isolated and propagated in culture media; however, no specific parameters endorse the quality of DSF inoculants. The main aim of the present research was to evaluate other plant variables that could be important for selecting efficient inocula for consortia propagation that mitigate the adverse effects of soil salinity. This is especially important because the efficiency of fungal isolates is not always related to the degree of mycorrhizal colonization [63] or the number of spores, which are the parameters commonly used to select and identify the quality of AMF inoculants.

#### Effectiveness of Fungal Consortia Propagated on Alfalfa

Previous studies showed that Na concentrations higher than 60 mM negatively affect fresh weight in alfalfa [64,65,66]. Consortia 2, 3, 7, and 17 in the present study increased fresh weight even at higher soil Na concentrations, demonstrating their effectiveness.

Some research has shown that soil Na concentration higher than 50 mM negatively affects N absorption [67,68,69]. In the present research, alfalfa plants inoculated with Consortium 11 with soil as inoculum had the highest foliar N content (4.6%) despite the Na soil concentration (181 mM) that generated this consortium (Table 1). Similarly, root-inoculated plants of Consortia 4, 6, and 16 had the highest foliar N values (3.7%, 3.1% and 3.7%, respectively). According to Mattson [70], foliar N content between 3% and 7% is typical, but values between 0.5% and 1.5% are low. Non-inoculated plants and inoculated with some consortia had low foliar N concentrations (Figure 6a).

Soil salinity is known to also decrease soil P availability [71] and inhibit its absorption by plants [72,73]. Most of consortia, using soil or roots as inoculum, positively influenced greater P foliar concentration in alfalfa plants. According to Malhotra et al. [74], normal foliar P range from 0.5 to 5 g P kg^−1^ DW. Consortia 1 and 3, using soil as inoculum, and Consortia 8 and 10, using roots, all come from soils with high Na concentrations, increased P absorption in alfalfa plants more than in non-inoculated plants (Figure 6b). Better P plant uptake in mycorrhizal plants is a well-known effect of mycorrhizal inoculation. AMF consortia from saline soils can increase foliar P concentration in sorghum when used as a trap culture [75]. DSFs offer similar benefits to those known in AMF. For example, they enhance P absorption in their host plants [12] and mitigate the effects of salt stress [60,61]. Therefore, in the present research, AMF and DSF participated in the nutritional status of alfalfa plants inoculated with consortia from two inoculum types. It can be hypothesized that each fungal type participates in the alfalfa mineral nutrition through different mechanisms. Della Monica et al. [76] observed a close relationship of AMF and DSF with P availability and absorption in plants. While DSF increases the rhizosphere P reserve, AMF transfer P to the host plant. Co-colonization of plants by AMF and DSF may have a synergistic effect. Future research should address this hypothesis to understand plant nutrition in saline soils, considering the dual participation of these two beneficial endophytic fungi.

Results from this research evidenced that several consortia (9, 12, and 15), but in particular Consortia 1 and 4 (Figure 7a), reduced the Na uptake by alfalfa even at the high soil Na concentration observed in the original soils used for their propagation (Table 1). Moreover, results support the knowledge that salinity-adapted consortia can control Na absorption in inoculated plants [77,78]. The Na soil concentration was almost three times more Na (655.9 mM) than the experiments conducted by Campanelli et al. [67] and Ben-Laouane et al. [79] with 150 mM and 120 mM NaCl saline solution in alfalfa plants, respectively. However, Campanelli et al. [67] quantified eight times higher foliar Na concentration (1447 mmol kg^−1^ DW) in plants inoculated with *Gl. viscosum* than the lowest Na foliar concentration (168 mmol kg^−1^) in the present study with native consortia. Ben-Laouane et al. [79] observed a Na foliar concentration of 870 mmol kg^−1^ in alfalfa plants inoculated with an AMF consortium (*Glomus* sp*., Sclerocystis* sp*.,* and *Acaulospora* sp.). Both studies reported higher concentrations than those observed in the present investigation in saline soils with native microorganisms from halophilic plants. However, it is important to mention that such the already mentioned studies were conducted in greenhouse conditions using non-saline soil (substrate) and that Na source (NaCl) induced salt stress. Some researchers have demonstrated that salinity-adapted consortia can control Na absorption in inoculated plants [77,78]. Estrada et al. [80] observed different behavior between AMF from saline and non-saline soils. In their work, corn plants irrigated with 0.1 M NaCl previously inoculated with *R. irregularis* from non-saline soil or *Claroideoglomus etunicatum* from saline soil. These authors reported that plants inoculated with *R. irregularis* showed a foliar Na concentration of 521 mmol kg^−1^ DW, but plants inoculated with *Claroideoglomus etunicatum* (adapted to salinity) had 260 mmol kg^−1^ DW, and non-inoculated plants had 869 mmol kg^−1^ DW of foliar Na. AMF decreased Na absorption in both inoculated plants, but absorption was more efficient in the saline soil isolate. Therefore, results obtained in the present research enable the selection of native halophilic endorhizal consortia that significantly affect salinity mitigation, as measured from the reduced foliar absorption of Na. In the same context, stress tolerance is a prerequisite for a successful symbiotic relationship between DSF and host plants that decreases salt stress. Therefore, fungi adapted to saline conditions will confer tolerance compared to non-adapted fungi [78]. The DSF *Curvularia* sp. isolated from the halophyte plant *Suaeda salsa* established a beneficial symbiosis with white poplar (*Populus tomentosa*), increasing its response to salt stress through higher antioxidant activity [77]. In the present investigation, native DSF associated with halophyte plants could confer tolerance to abiotic stress in alfalfa, but separating their effects from those of AMF is impossible. Therefore, specific studies are necessary to test the independent efficiency of these endophytic fungi.

Protective osmolytes (K, Ca, and Mg) are key in mitigating plant salt stress because they can act as osmotic adjusters. Na ions compete with K ions for binding sites essential for several cellular functions [4]. Research shows that inoculation with AMF favors K absorption over Na absorption in saline or sodic soils. In the present study, the consortia from the soil, except the 15, improved foliar K concentration in alfalfa. In particular, five consortia (1, 8, 17, 18, and 19) enhanced it more than three-fold than in control plants. The results show that although the soils from Consortia 1 and 8 had more Na than K, the plants absorbed preferentially more K than Na (Figure 7b). Inherently, the consortia from roots also improved K uptake in alfalfa plants. This is in accordance with the studies of Zuccarini and Okurowska [81]. These authors observed higher leaf K concentration (1069 mmol kg^−1^ DW) in plants of sweet basil inoculated with a mycorrhizal consortium composed of *Gl. mosseae* (now *Funneliformis mosseae*), *R. irregularis,* and *F. coronatum* than in control plants (820 mmol kg^−1^) when irrigated with 50 mM NaCl. However, it is not always the case. These authors found no differences at higher NaCl concentrations (250 mM) or in the association *R. irregularis*–*Ocimum basilicum* with 50 mM NaCl. Therefore, the results may differ depending on the fungi and the Na:K ion ratio.

The present data suggest that AMF and the native DSF communities in the consortia participated in a significant K increase in alfalfa plants (Figure 7b). Yun et al. [82] reported that *Zea mays* plants irrigated with 200 mM NaCl and inoculated with *Piriformospora indica* had higher foliar K concentrations than their non-inoculated counterparts. Similar results were reported by Song et al. [83] in *Hordeum vulgare* inoculated with *Epichloe* sp. and irrigated with 200 mM NaCl, and inoculated plants had higher leaf K concentration (512 mmol kg^−1^ DW) than the control (385 mmol kg^−1^ DW). In contrast, Ghabooli [84] observed no differences in leaf K concentration between control plants of *H. vulgare* and those inoculated with *P. indica*, both with 200 mM irrigation. Therefore, the type of fungus may have a relevant effect on the absorption of K, an osmoregulatory element.

According to LaHaye and Epstein [85], soil Na concentration greater than 50 mM affects Ca uptake in plants. In the present research, the original soil of all the consortia, except 16, 18, and 19, exceeded this saline soil concentration but substantially increased foliar Ca concentration in alfalfa plants. It is known that AMF can favor Ca absorption in their host plants under saline conditions. For example, Evelin et al. [68] reported differences in leaf Ca concentration between *Trigonella foenum-graecum* plants inoculated with *Gl. intraradices* (115 mmol kg^−1^) and the non-inoculated ones (95 mmol kg^−1^ DW), both irrigated with 200 mM NaCl. In the present research, plants inoculated with Consortia 1, 2, and 3, with very high Na concentration in the soil of origin, had 2.7 times higher Ca concentration than control plants. However, Consortia 16 and 19 from soil were the ones that increased the foliar Ca concentration the most (998 and 1040.67 mmol kg^−1^ DW, respectively). Marschner [86] indicated that the optimum range of foliar Ca concentration in plants is between 25–1200 mmol kg^−1^ DW.

It is known that DSF increase leaf Ca concentration under salinity conditions. For example, in *H. vulgare* inoculated with *P. indica* and irrigated with 300 mM NaCl, the foliar Ca concentration was 299 mmol kg^−1^ DW, and in non-inoculated plants, it was 155 mmol kg^−1^ DW [84]. In *Lolium arundinaceum* plants irrigated with 250 mM NaCl, inoculated and not inoculated with DSF, the leaf concentration was 64 mmol kg^−1^ DW and 52 mmol kg^−1^ DW, respectively [87].

Research has shown that AMF can increase leaf Mg concentration under saline [88] and non-saline conditions [89], although this is not always the case. Giri and Mukerji [88] inoculated *Sesbania aegyptiaca* and *S. grandiflora* plants with *Gl. macrocarpum* isolated from saline soil (15 dS m^−1^ EC and 150 mM NaCl) and reported 67 mmol kg^−1^ DW in non-mycorrhizal plants and 134 mmol kg^−1^ DW in mycorrhizal plants. Evelin et al. [68] found no differences in leaf Mg concentration between inoculated and non-inoculated plants in *T. foenum-graecum* plants inoculated with *Gl. intraradices* and irrigated with 200 mM NaCl. In the present research, all inocula, from soil or roots, significantly increased foliar Mg concentration in alfalfa plants. It was 5.6 times higher than the maximum value obtained with Consortium 13 (using soil as inoculum) and 7.4 times with Consortium 9 (using roots), both in comparison to non-inoculated plants that had foliar Mg concentration below the normal foliar Mg concentration (between 61 to 205 mmol kg^−1^), according to Marschner and Cakmak [86]. DSF also improve leaf Mg concentration, which has only been shown in non-saline conditions. For example, in tomato plants inoculated individually with DSF (A101, A103, and A105), the average leaf Mg concentration was three mmol kg^−1^ DW, while in non-inoculated plants, the concentration of Mg was 2.5 mmol kg^−1^ DW [90]. In contrast, rice crop showed a significant difference in leaf Mg concentration between plants inoculated with DSF (A101 and A103) and non-inoculated plants [10]. Further research should describe the individual and combined functions of AMF and DSF on nutrient transfer, and the functionality of AMF and DSF under saline conditions should be integrated. Several studies with AMF have ignored the natural participation of DSF, and sometimes they even were reported as contaminants [91]. The participation of endorhizal fungi in increasing macronutrients with osmoprotector activity is relevant to decrease salinity stress. Plants established in sodic soil can stress due to macronutrient deficiencies of Ca, Mg, and K [92].

### 4.4. Scanning Electron Microscopy (SEM) and Elemental Analysis in AMF Fungal Structures

Similar to the results obtained in the present research, Hammer et al. [5], using PIXE analysis, observed low Na content in spores and hyphae from two natural saline sites with different Na concentrations. The result obtained in the present research clarifies that the lower amount of Na in alfalfa tissues is not due to the involvement of these fungal structures in Na accumulation but that other mitigation mechanisms to control salt uptake might be participating. Although the soil Na concentrations in the present research were up to three times higher than those found by Hammer et al. [5], the structures from AMF did not accumulate high concentrations of Na (0.03–0.29%), K (0.13–0.15%), and Mg (0.96%) or Ca (2.3%). These authors found up to 0.49% of Na, 0.89% of K, 0.96% of Mg, and 12% of Ca in fungal structures in soil containing 200 mM of Na. The higher presence of Ca over Na in AMF structures implies, like in plants [4], that Ca acts as an osmotic element in fungal tissues, demonstrating the ion-selective uptake capacity of AMF. However, little is known about the osmotic mechanisms by which AMF tolerates salinity conditions and how other ions participate in the process that mitigates Na toxicity.

Si was also detected in fungal structures propagated from the two types of inoculum analyzed. Si is a common element present in the soil. Still, the presence of Si in the spores and hyphae of Consortium 11 is relevant because the inoculum source, i.e., the root, produced the structures containing this element, suggesting that Si is inside the cytoplasm of AMF from saline soils. Hammer et al. [5] reported Si content of 1% in spores. Si addition has been proposed as an alternative to increasing plant resilience to salinity because it modulates physiological and biochemical processes altered by salt stress. However, the molecular mechanisms are still being studied [93]. Under salinity conditions, AMF enhance Si absorption in plants and induce Na precipitation in cell walls of root cells [94], thereby reducing Na concentration in shoots [95]. The mechanisms by which AMF participates in Si absorption are still unclear, so further study is needed to understand their function and the involvement of DSFs in this process.

### 4.5. Principal Component Analysis (PCA)

PCA (Figure 9) demonstrated that the physiological plant effects analyzed in this research related to salinity stress were essential in selecting the most promising fungal inoculum. Fungal infectivity (colonization percentage and the number of spores) and their effectivity (foliar N, P, K contents) significantly influenced plant fitness in the trap alfalfa cultures established from the 38 samples (from soil and root) analyzed. The present research is the first to show that effectivity is a relevant parameter in selecting native saline-tolerant endorhizal fungi that may be useful to mitigate salinity stress. Moreover, both kinds of fungi (AMF and DSF) were important and developed functions in alfalfa plants that did not overlap, but they were complementary to each other. Routsalainen et al. [15] reported that plants benefited from DSF but more from AMF. The present research demonstrated a more substantial relation of AMF with several physiological plant variables than DSF. Therefore, endorhizal effectiveness is a quality trait to select fungal salinity-adapted consortia useful as plant inoculants with a high possibility for mitigating plant stress under salinity conditions.

The present investigation also indicates the efficiency of the fungal Consortia 1, 3, and 10 native to saline soils and associated naturally with halophyte plants, which promote P absorption in alfalfa plants when using the two inoculum tested. Therefore, P absorption should be consistently evaluated as a measure of the effectiveness of AMF in saline soils. The AMF was also closely related to K. Both P and K foliar concentrations were important variables in PC1. Several studies have shown that AMF improves P absorption in plants [96]. Plants better nourished with P and K will be more tolerant to salinity stress.

PCA showed no relationship between AMF and DSF colonization. However, studies should be conducted to better understand the functioning of these endorhizal fungi on their host plants under salt-stress conditions. This will promote their beneficial use in saline soils.

### 4.6. Selection of Fungal Inoculants

Consortium 1 was selected by several infectivity and effectivity variables. Although several consortia had a percentage of AMF colonization within the range (43%) proposed by INVAM [17] for inoculant selection, other relevant variables were useful for selecting this consortium. A remark is that this information should not be regarded to formulate a fungal inoculum but enforce the scientific knowledge of fungal strains native from halophyte plants and propagated in alfalfa as hosts in trap cultures with physiological plant benefits to face salinity stress. Hence, effectiveness is also helpful for selecting AMF/DSF inoculants and should be implemented as part of plant variables qualifying the inoculum quality. Abbott et al. [97] postulated that inoculant effectiveness is based on improving plant P absorption. However, the point must also be based on absorbing other vital plant and soil nutrients (N and K) that are related to particular soil conditions, such as high salinity. Results demonstrated that the selection of inoculants from saline soils should be based on physiological processes (nutrient absorption) that highlight the intrinsic abilities of AMF/DSF and generate benefits for the plant. Plants inoculated with this consortium had the lowest foliar Na concentration (180 mmol kg^−1^) despite having the highest Na concentration in the original soil (656 mM), i.e., it had four times more Na than the leaf tissue. It also effectively increased foliar K concentration despite the high Na concentration in the soil. Although the soil P concentration of Consortium 1 was below the average soil concentration, the fungi from Consortium 1 favored P uptake. Finally, physiological plant traits modified by endorhizal fungi are outstanding variables under saline conditions that also evidence the potential use of these fungi for mitigating the harmful effect of salinity in plants.

### 4.7. Identification of AMF Spore Morphospecies in the Selected Consortium

The morphospecies of Consortium 1, after fungal trap culture in alfalfa, did not change composition from the original soil sample that originated in this consortium. Instead, it observed changes in their percentage. Some of the fungal morphotypes in this consortium were observed earlier in saline soils. For example, Wang et al. [98] mentioned that the AMF commonly observed in saline soils are *Glomus* sp. Aliasgharzadeh et al. [53] observed that the predominant AMF species in salty soils with EC of 162 dS m^−1^ were *Gl. intraradices* (now *R. irregularis*), *Gl. versiforme* and *Gl. etunicatum* (now *C. etunicatum*). Bencherif et al. [26] observed 12 AMF species after establishing a trap culture with alfalfa by using soil as inoculum at three EC levels. *Dominika* sp. was the dominant mycorrhizal species in the highly saline soil with EC of 9.9 dS m^−1^, whereas *R. irregularis* was the dominant mycorrhizal species in the medium saline soil with 8.5 dS m^−1^ and saline soil with 4.1 dS m^−1^. Only *R. irregularis* was the common morphospecies found in these three studies. Further research is being conducted to molecularly identify native AMF and DSF species that colonize the roots and soil of halophyte plants and those that were propagated in alfalfa trap cultures.

### 4.8. Endorhizal Community of the Selected Consortium

The dominance of Ascomycota and Glomeromycota is in accordance with the endorhizal fungi studied. Currently, all DSF described belong to the phyla Ascomycota [99]. Sequencing results corroborated the presence of DSF in *D. spicata* roots. In addition, the genera *Fusarium* and *Preussia* have been classified as DSF, and some species have even been reported as plant growth promoters [91,100]. Although the molecular analysis identified the genus *Rhizophagus*, it did not detect the other three morphotypes mentioned previously as components of the Consortia 1. Therefore, more specific molecular analyses for AMF should be performed to molecularly identify the morphospecies observed in Consortium 1. Few reports have reported the composition of the endophytic fungal community of *D. spicata*. Redman et al. [101] informed that DSF of the genera *Phoma* sp. and *Fusarium* sp. were the most dominant in the roots of *D. spicata*. Other studies also agree that the dominant phyla in some halophyte plants (*Inula crithmoides*) is Ascomycota, and the genus *Fusarium* (50%) is the most predominant [102]. Of note, *Fusarium* is a very diverse fungal group that includes pathogenic or saprotrophic species. *Fusarium* did not cause damage in alfalfa roots in the fungal trap culture; however, its effect should be analyzed before its wide use in other plants. These results expand the knowledge of the components that may be related to the quality of the endorhizal fungal inoculum produced. The bacterial community was outside of the scope of this research. However, it should be molecularly analyzed as these microorganisms are also a primordial microbial component of saline soils and form part of inoculants for sustainable agriculture.

## 5. Conclusions

Traditionally, fungal type, mycorrhizal colonization, and spore number determine the infectivity of fungal inoculants used for commercial and research purposes. It is the first study addressing the selection of native endorhizal fungi adapted to high-salinity conditions, using soil and roots of halophyte plants as sources of fungal inoculum. This research also shows that under salt stress conditions, the foliar K, Ca, Mg (protective osmolytes that mitigate salinity damage), N, and P (essential nutrients) concentrations can be indicators for the selection of inoculants that effectively increase plant tolerance under high salinity conditions. In this context, the results provide further information on variables useful for selecting native inocula from saline soils that participate in the mitigation of plant salinity. Moreover, they may have the potential for their use in the sustainable management of agroecosystems. In particular, Consortium 1 improved the absorption of P and K, decreased the foliar concentration of Na in alfalfa, and produced the highest number of spores. The results offer a novel alternative endorhizal inoculants for selection that are adapted to salinity and are effective in protecting plants and probably increasing crop yields in saline soil conditions. Future research should analyze this in field conditions and other salinity tolerance mechanisms by which these endorhizal fungi protect their plants because it is a problem predicted to increase under the current global climate change scenario.

## Figures and Tables

**Figure 1 jof-09-00893-f001:**
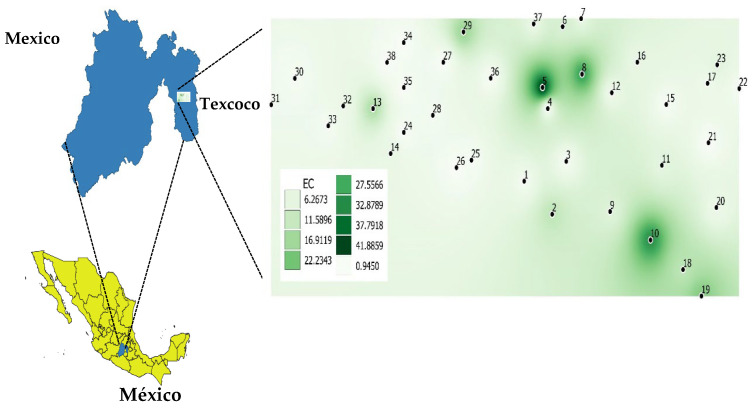
Spatial distribution of soil salinity and samples collected for this study. EC = electrical conductivity (dS m^−1^).

**Figure 2 jof-09-00893-f002:**
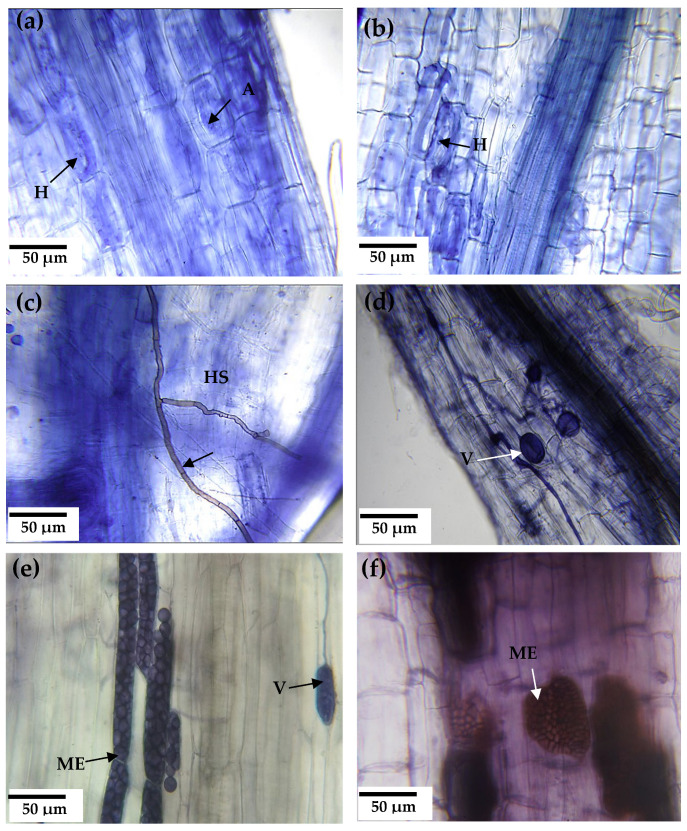
Endorhizal root colonization in alfalfa as host plant in trap cultures. Alfalfa inoculated with soil: (**a**) Consortium 1, (**b**) Consortium 3, and (**c**) Consortium 6. Alfalfa inoculated with roots: (**d**) Consortium 3, (**e**) Consortium 1, and (**f**) Consortium 7. Structures of arbuscular mycorrhizal fungi = arbuscules (A), vesicles (V), and hyphae (H). Structures of dark septate fungi = microsclerotium (ME) and dark septate hyphae (HS).

**Figure 3 jof-09-00893-f003:**
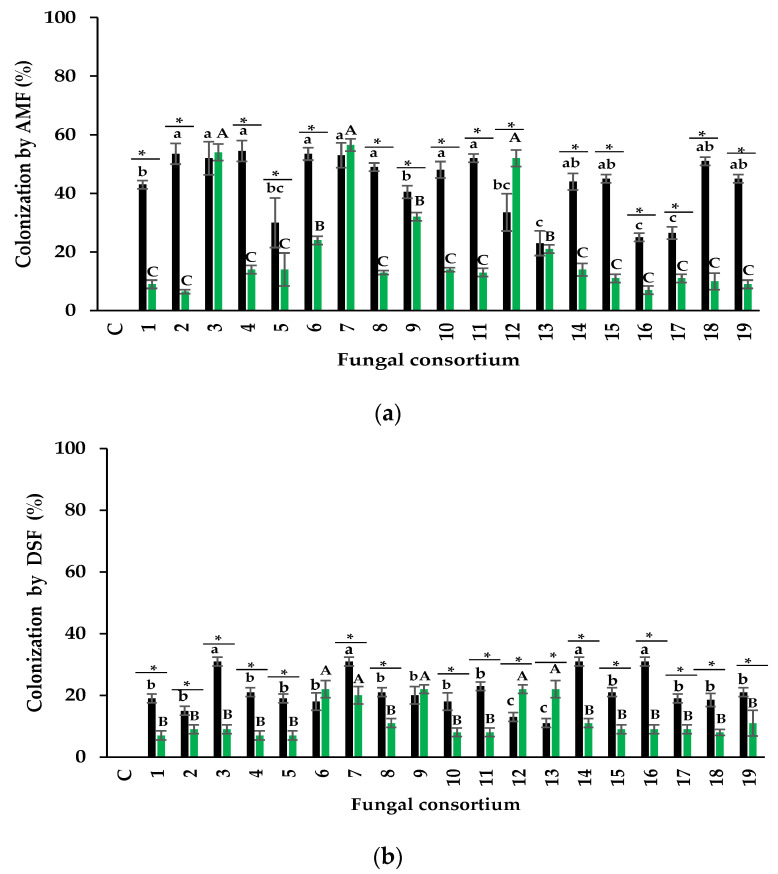
Colonization by arbuscular mycorrhizal fungi (**a**) and dark septate fungi (**b**). Black bars = soil as inoculum, and green bars = root as inoculum. Values correspond to average ± standard deviation, n = 3. Different lowercase letters show colonization differences when comparing fungal consortia using soil as inoculum. Different capital letters show colonization differences when comparing fungal consortia using roots as inoculum. * Show a significant difference in the percentage of colonization of each fungal consortium when comparing soil and root as inoculum (Tukey, α = 0.05).

**Figure 4 jof-09-00893-f004:**
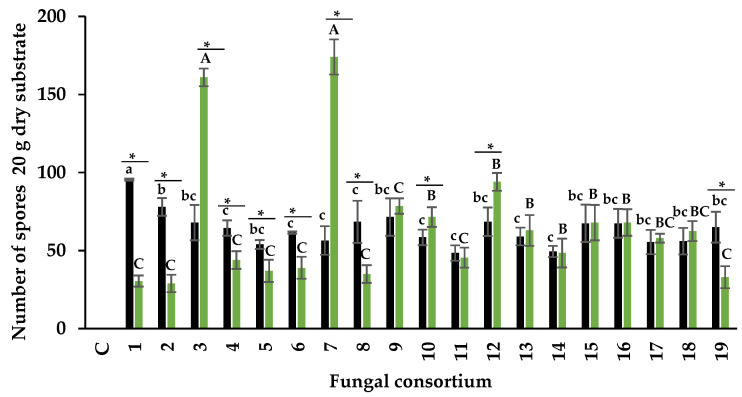
Number of spores of AMF in alfalfa trap cultures with two types of inoculum. Black bars = soil as inoculum, and green bars = root as inoculum. Values correspond to average ± standard deviation, n = 3. Different lowercase letters show spore number differences when comparing fungal consortia using soil as inoculum. Different capital letters show spore number differences when comparing fungal consortia using roots as inoculum. * Show a significant difference in the spore number of each fungal consortium when comparing soil and root inoculum (Tukey, α = 0.05).

**Figure 5 jof-09-00893-f005:**
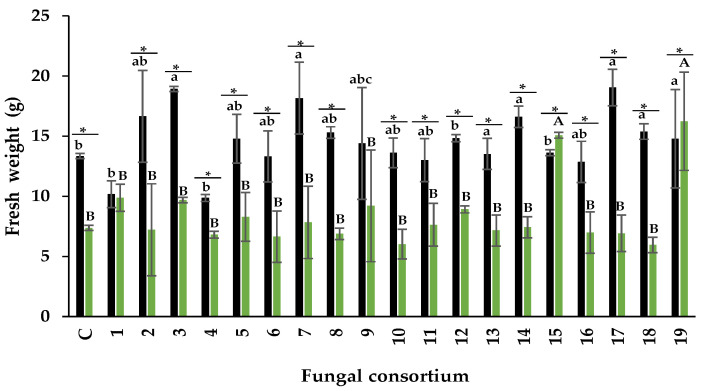
Fresh weight of the aerial part in alfalfa culture with two types of inoculum. Black bars = inoculum soil, and green bars = root inoculum. Values correspond to average ± standard deviation, n = 3. Different lowercase letters show differences in fresh weight when comparing fungal consortia using soil as inoculum. Capital letters show differences in fresh weight when comparing fungal consortia using roots as inoculum. * Show the significant difference in fresh weight of each fungal consortium when comparing soil and root inoculum (Tukey, α = 0.05).

**Figure 6 jof-09-00893-f006:**
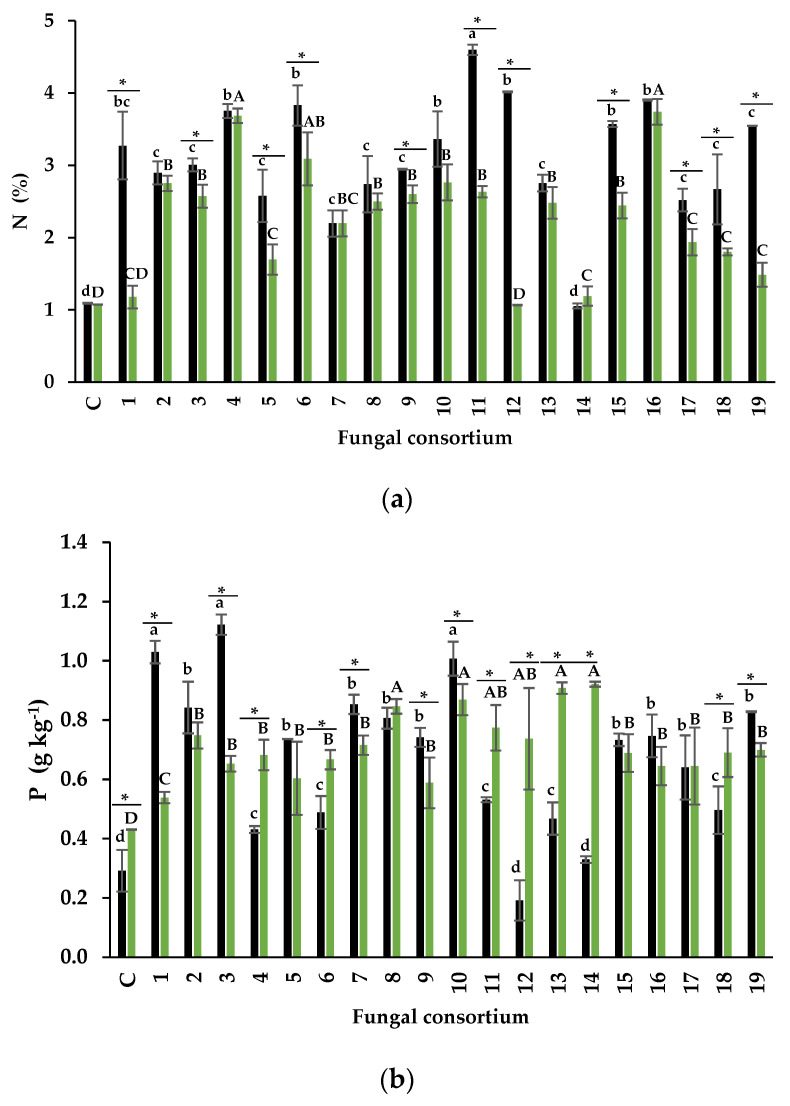
Influence of inoculation of fungal consortia on foliar nutrient concentration. (**a**) Nitrogen and (**b**) phosphorus in alfalfa trap culture with two sources of inoculum. Black bars = inoculum soil, and green bars = root inoculum. Values correspond to average ± standard deviation, n = 3. Different lowercase letters show nutrient concentration differences when comparing fungal consortia using soil as inoculum. Capital letters show nutrient concentration differences when comparing fungal consortia using roots as inoculum. * Show a significant difference in nutrient concentration of each fungal consortium when comparing soil and root inoculum (Tukey, α = 0.05).

**Figure 7 jof-09-00893-f007:**
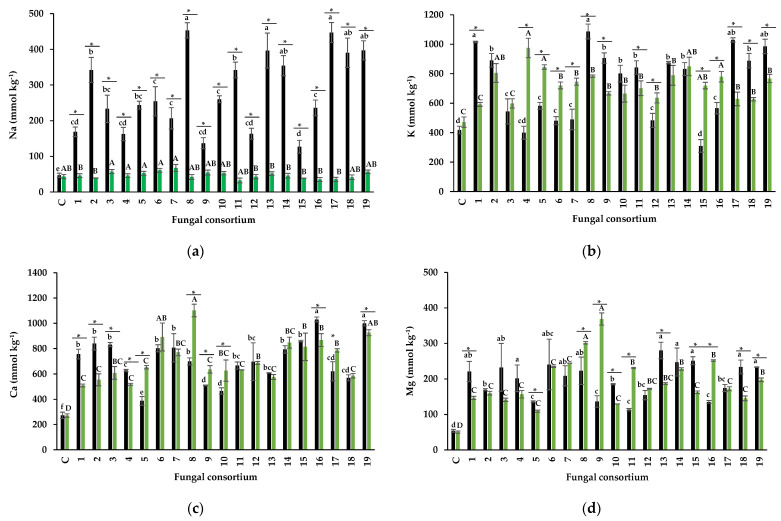
The Na foliar concentration and osmoprotective elements in alfalfa acting as fungal trap culture to propagate endophytic fungi. Black bars = inoculum soil, and green bars = root inoculum. Average ± standard deviation, n = 3. Different lowercase letters show foliar concentration differences when comparing fungal consortia using soil as inoculum. Different capital letters show foliar concentration differences when comparing fungal consortia using roots as inoculum. * Show a significant difference in foliar concentration of each fungal consortium when comparing soil and root inoculum (Tukey, α = 0.05). (**a**) Na, (**b**) K, (**c**) Ca, and (**d**) Mg concentration.

**Figure 8 jof-09-00893-f008:**
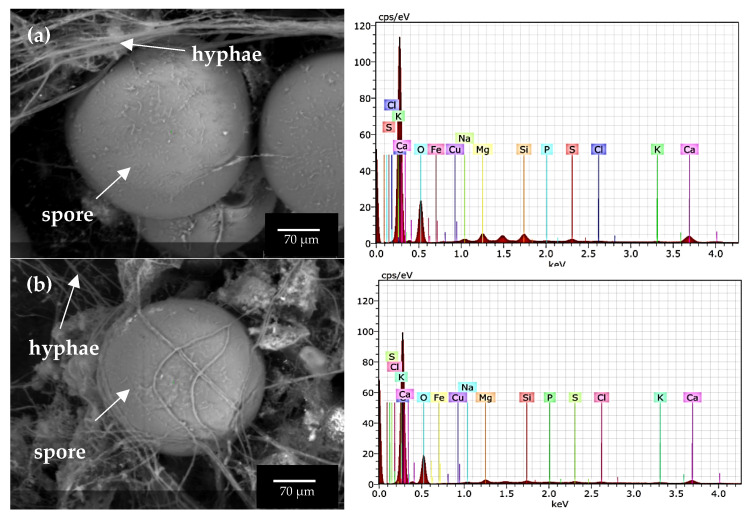
Micrographs acquired with an environmental electron microscopy of spores and hyphae from the alfalfa trap culture and the elemental content obtained using EDX. (**a**) Consortium 2 and (**b**) Consortium 8, both from soil used as inoculum in trap cultures.

**Figure 9 jof-09-00893-f009:**
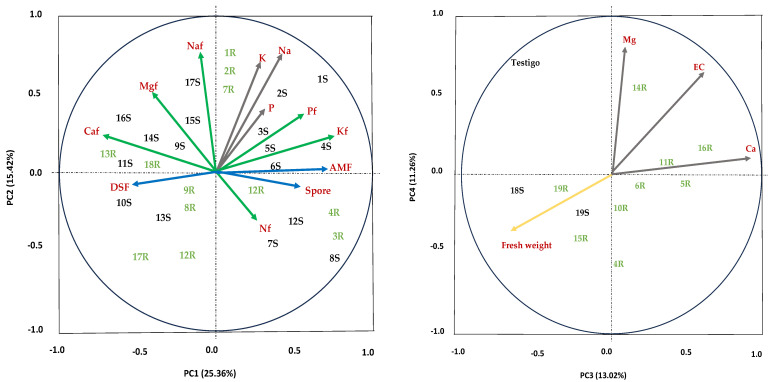
Principal component analysis of data of infectivity and effectivity responses in alfalfa plants used as fungal trap culture inoculated with soil or roots from halophytes established in saline soils. Variables related to fungi (blue arrows) and plants (green and yellow arrows) and original soil variables (gray arrows). AMF = colonization by arbuscular mycorrhizal fungi or by dark septate fungi (DSF); Concentration of elements in soil = chemical symbols or foliar alfalfa concentration = chemical symbols with f, and EC = Electrical conductivity. The numbers refer to a consortium from soil (black color) or from roots (green color).

**Figure 10 jof-09-00893-f010:**
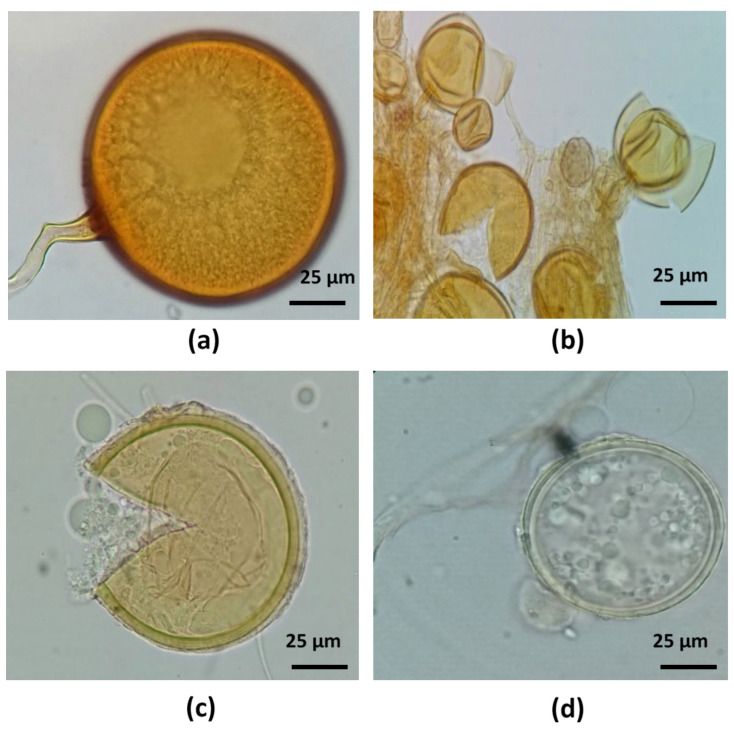
Arbuscular mycorrhizal fungi collected after and before soil Consortium 1 trap culture in alfalfa. (**a**) *Funneliformis mosseae*, (**b**) *Rhizophagus aggregatus*, (**c**) *Claroideoglomus claroideum*, and (**d**) *Diversispora* sp.

**Figure 11 jof-09-00893-f011:**
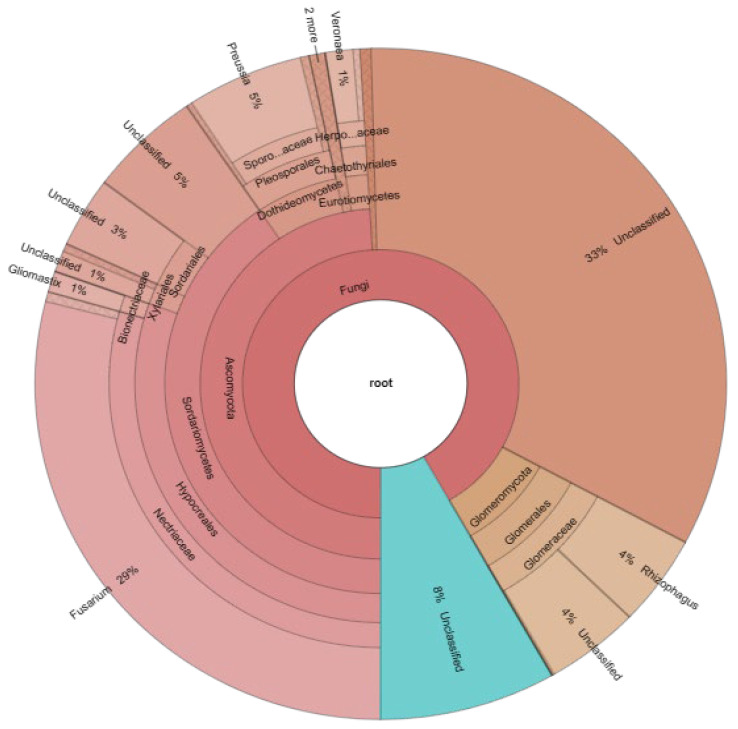
Analysis of the taxonomical annotation for endophytic fungi found in *Distichlis spicata*, where Consortium 1 originated.

**Table 1 jof-09-00893-t001:** Cations and phosphorus concentration, electrical conductivity (EC), pH, and Na adsorption ratio (SAR) in soil compound samples from halophyte species in saline soils used for endorhizal fungal trap cultures.

Consortium	# of Individual Soil Samples ^†^	Cations Concentration (mM) ^††^				EC (dS m^−1^) ^††^	pH ^††^	SAR (mM) ^††^	Soil Type		PO_4_^−2^ Olsen (mg kg^−1^) ^††^	Vegetal Species
		Na	K	Ca	Mg				S	SS		
1	5	655.9	60.2	2.0	0.7	42.5	7.8	580.4		+	11.0	*Distichlis spicata*
2	8, 10	585.3	70.8	5.1	1.4	36.3	8.1	325.1		+	15.5	*Distichlis spicata*
3	19	322.9	39.3	3.3	0.8	20.3	8.1	225.1		+	12.2	*Kochia scoparia*
4	29	272.5	15.2	3.0	0.5	17.5	9.2	206.4		+	15.1	*Eragrostis obtusiflora*
5	13	222.0	18.3	3.3	0.8	15.4	8.3	154.0		+	13.9	*Distichlis spicata*
6	2, 18	212.0	24.7	3.3	2.4	14.8	8.5	213.6		+	15.7	*Suaeda torreyana*
7	6	94.9	12.6	2.3	1.5	8.5	8.1	69.2		+	13.3	*Distichlis spicata*
8	9, 11	131.2	18.6	4.4	3.1	9.1	8.4	67.2		+	16.5	*Baccharis salicifolia*
9	20, 23	52.4	16.8	8.2	2.1	8.7	8.6	23.0		+	13.0	*Distichlis spicata*
10	7	80.8	9.1	3.9	1.6	6.0	9.4	49.0		+	17.4	*Kochia scoparia*
11	12, 14, 16	181.6	12.3	5.7	3.2	6.4	9.6	86.5		+	15.0	*Eragrostis obtusiflora*
12	32, 37	212.0	12.6	3.1	1.7	6.6	8.2	135.2		+	6.5	*Eragrostis obtusiflora*
13	17	50.5	9.1	3.9	1.0	3.8	8.5	32.5	+		8.1	*Cynodon dactylon*
14	4	52.5	8.1	4.9	1.4	3.1	8.7	29.6	+		10.9	*Baccharis salicifolia*
15	3	131.1	8.1	6.0	1.4	2.6	8.7	68.3	+		16.1	*Kochia scoparia*
16	21, 22	38.3	7.3	3.9	3.0	2.2	8.4	20.6	+		10.8	*Distichlis spicata*
17	1, 27, 28, 30, 31, 33	141.3	3.4	3.1	1.2	3.8	7.9	96.8	+		7.2	*Eragrostis obtusiflora*
18	15, 34, 35, 36, 38	31.3	6.3	6.3	3.3	2.0	8.0	14.3	+		8.8	*Baccharis salicifolia*
19	24, 25	31.3	6.0	5.2	1.6	0.9	8.4	17.0	+		12.1	*Kochia scoparia*
Normal in the soil solution		0.2 ^†††^	20 ^††††^	0.1 ^‡^	0.12–8.5 ^‡‡^							

^†^ Individual soil samples (38) from the rhizosphere halophytes established in saline (S) or saline-sodic (SS) soil. ^††^ Values correspond to the final sample used in the fungal trap cultures (consortium). ^†††^ Data according to Rodríguez-Navarro [29], ^††††^ Borax [30], ^‡^ Wood et al. [31], ^‡‡^ Marschner, 2012 [32].

## Data Availability

All generated information is presented in this document.

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
