# Peer review of "Selection of Salinity-Adapted Endorhizal Fungal Consortia from Two Inoculum Sources and Six Halophyte Plants"

_jof, 2023, doi:10.3390/jof9090893_

Round 1

Reviewer 1 Report

Review: J-Fungi 2023, 9, x. https//doi.org/10.3390/xxxxx

General consideration

The inoculums are considered for their ability to form arbuscular mycorrhizae and to be colonized by DSF. This leaves a vacuum at the bacteria level. The choice to consider this manuscript for publication is, in my opinion, up to the editor, but the absence of in deep analysis of the bacterial compartment is a major bias for this study device.

Moreover, the technical itinerary for the production of inoculum with a host plant for an amplification phase of the natural inoculum makes the experiment very difficult to reproduce and even if the material and method describe these points very well, the formulation of the inoculum is more empirical than scientific.

Soil salinization

When we talk about salinization, we must specify which salts we are talking about. In some cases, salinization is reduced to NaCl and in others it is extended to other salts: Ca, Na, K and Mg with CO3, SO4, Cl in particular. Please clearly specify.

2.1- the sampling site occupies a very small area at the edge of a particularly important and active agricultural environment. Under these conditions, the sampling site is necessarily and strongly impacted by the neighboring agricultural environment: presence of weeds, fertilizers, pesticides, etc. This environment and the sampling site need to be better described.

2.5.1- Segments were clarified with 3% KOH for 12 days until the removal of root pigmentation.

12 days: are you sure of that?

Trypan, not trypan.

2.6- The processing of sequences until their taxonomic assignment on databases is not sufficiently described.

Results and discussion

The number of spores is a very difficult variable to interpret. In vitro, strictly under the same conditions, out of 100 dishes, 10 or 20 will not produce spores and the others very variable quantities. So, under less controlled conditions… The corresponding section needs reformulation to avoid over-interpretation.

Figure 8: the 3D representation is difficult to read. I prefer two side-by-side graphs with Dim1 and Dim2 and the second graph with Dim3 and Dim4.

3.7- What is the identification of fungi based on: spore morphology only or morphology and sequencing?

3.8- The description made of the bacterial community is useless. It was necessary to implement a differential description with interaction network analyses. Given what is presented, I propose to just indicate that it has been done, to put it in additional data if necessary.

English easy to understand and clear writing style

Reviewer 2 Report

1. the language is a little bit difficult to be understand well. As a scientific directly descript is necessary for the meaning explains. 2.in introduction, the interaction of DSF and AMF could be explained with references, ie. the other researchers` results related to DSF; 3. more information of DSF needs to be added, including the differences between DSF and AMF. 4. results and discussion could be separated in order to give out a clear information between the present study and other`s work. 5. the figs could be bigger, and some letters in fig are overlapped. 6.the support information could be added in materials and methods. Finial, the main aim and main result could be clearly pointed out. 1. the language is a little bit difficult to be understand well. As a scientific directly descript is necessary for the meaning explains. 2.in introduction, the interaction of DSF and AMF could be explained with references, ie. the other researchers` results related to DSF; 3. more information of DSF needs to be added, including the differences between DSF and AMF. 4. results and discussion could be separated in order to give out a clear information between the present study and other`s work. 5. the figs could be bigger, and some letters in fig are overlapped. 6.the support information could be added in materials and methods. Finial, the main aim and main result could be clearly pointed out.

Round 2

Reviewer 2 Report

please make sure the pictures are clear enough for readers.